



# Retrieval of Intensive Aerosol Microphysical Parameters from Multiwavelength Raman/HSRL Lidar: Feasibility Study with Artificial Neural Networks

**M. Mustafa Mamun[1], Detlef Müller[2,*]**

[1] Gwangju Institute of Science and Technology, Gwangju, South Korea
[2] University of Hertfordshire, Hatfield, United Kingdom.
[*] Correspondence to: Detlef Müller (d.mueller@herts.ac.uk)

**Abstract.** We present results of a feasibility study that uses Artificial Neural Networks (ANN) for the retrieval of intensive microphysical parameters of atmospheric pollution from combinations of backscatter ($\beta$) and extinction coefficients ($\alpha$) that can be measured with multiwavelength Raman and high-spectral resolution lidar at 355, 532, and 1064 nm. We investigated particle effective radius, and the real and imaginary part of the complex refractive index. ANN could be a useful alternative or

supplementary method over the traditional approach of retrieving microphysical particle properties with classical inversion algorithms because data analysis with ANN is significantly faster and allows for investigating the information content of the optical input data. We investigated the data combinations $3\beta+2\alpha$, $3\beta+1\alpha$ (355 and or 532 nm), $2\beta$ (532, 1064 nm) $+1\alpha$ (532 nm), and $3\beta$ with Feedforward Backpropagation Multilayer Perceptron Neural Networks. The synthetic optical data were computed

with a Mie-scattering algorithm for monomodal particle size distributions. Mean radii of the size distributions ranged between 0.01 and 0.5 µm, and mode widths ranged between 1.4 and 2.5 resulting in effective radii between 0.13 and 4.1 µm. We tested real parts between 1.2 and 2, and imaginary parts between 0.0$i$ and 0.1$i$. The complexity of developing the networks did not allow us to test the influence of measurement errors of the optical data but the error produced by the ANN can be quantified. From

the five basic data combinations, our current network design allows us to derive effective radius with an accuracy of approximately ±16 to ±35%, and ±17 to ±39% if the true mean radii is in the range from



110 - 250 nm, and 260 - 500 nm, respectively. The real part can be derived with an accuracy of approximately ±7 to ±10%. We find retrieval errors of approximately ± 31 to ±38% for the imaginary part. We show that ANN can potentially estimate some particle parameters with various levels of uncertainty not only from what we denote as 3β+2α information but also from data combinations of 3β+1α (355 or 532), 2β (532, 1064) +1α (532), and 3β. We hypothesize that the ANN carries out first a pre-selections of various values of extinction-based Ångström exponents with regard to effective radius and then uses this information to create the strong correlation between particle effective radius and lidar ratios in all particle size distributions (PSDs) we investigated.

## 1    Introduction

Radiometers and light detection and ranging (LIDAR) instruments on satellites and at ground are fundamental methods for the investigation of the impact of atmospheric pollution and trace gases on global climate change. Reports regularly published by the Intergovernmental Panel on Climate Change, IPCC (IPCC and Press, 2013) point to the uncertainty of climate change forecast with regard to particulate pollution. Particulate pollution stems from natural and man-made sources, i.e., sea salt and mineral dust, smoke from forest fires, urban haze from traffic and industrial activities. These particles have different lifetimes that can span hours to weeks. They appear in various heights of the atmosphere from the ground to stratospheric heights. Their chemical and physical properties are variable and depend on generation mechanisms, transport time, and can be affected by various factors such as ambient humidity, the condensation of gases on existing particles, gas-to-particle transformations, and the mixing of particles in terms of internal and external mixtures. All these factors describe only a part of the complex aerosol system. Various experimental and theoretical methods are needed for the detection and characterization of these particles.

In the past 20 years Raman lidar has evolved into a standard method for the observation of aerosols (Ansmann and Müller, 2005). Lidar is the only method that allows us to detect particle properties under ambient atmospheric conditions on a vertically highly-resolved scale. These days many lidars can



operate under 24/7 conditions. Though aircraft missions can also provide profiles of aerosol profiles, these activities are limited to episodes during field campaigns. The time spans of data acquisition are by

far less than what can be achieved by, e.g., lidar which can measure for days or weeks unless hardware failure, maintenance, or adverse weather conditions force the shut-down of the instrument.

Lidar networks such as the European Aerosol Research Lidar Network (EARLINET), www.earlinet.org (Pappalardo et al., 2014) and space-borne lidars, of which CALIOP (Winker et al., 2004; Winker et al., 2007) aboard CALIPSO (Winker et al., 2009) is the first example of a successful

long-term aerosol lidar mission, have started to create an enormous amount of data that needs to be analyzed. Upcoming missions involve ESA's high-spectral resolution lidars aboard the ADM Aeolus (Stoffelen et al., 2005) and the EarthCARE (https://earth.esa.int/web/guest/missions/esa-future-missions/earthcare) satellites. Even though satellite missions employing passive remote sensors also generate a large amount of data the addition of the vertical resolution adds another dimension in the data

set. This spatial dimension leads to a significant increase of the amount and complexity of the data which should preferably be analyzed in real-time particularly with respect to space-borne lidar.

Since the early 1990s Raman lidar (Ansmann et al., 1990) has become the workhorse for quantitative aerosol characterization, as this lidar method allows for measuring particle backscatter and extinction coefficients. EARLINET activities in the past 15 years focused on developing infrastructure

in terms of reliable, quality assured instruments and software for data analysis. Automated-algorithms that allow for fast, unsupervised analysis of lidar signals acquired by the various lidar stations are in the test phase. This single-chain approach currently focusses on the retrieval of the optical aerosol parameters that can be measured with Raman lidar, notably backscatter and extinction coefficients. This approach follows the philosophy of coherent, fast analysis of signals acquired by the AERONET sun

photometer network (Holben et al., 1998; Holben et al., 2001).

The development of multiwavelength Raman lidar in the mid-1990s offered another dimension in exploiting Raman lidar data. Inversion algorithms (Kolgotin and Müller, 2008; Qing et al., 1989; Tikhonov, 1977; Twomey, 1977) have been developed with the purpose of extracting microphysical





information, such as particle size and the complex refractive index, from which optical properties such
as single scattering albedo can be inferred (Böckmann et al., 2005; Müller et al., 1998; Veselovskii et
al., 2002). Significant progress has been made with regard to inversion algorithm development. Case
studies dealt with the proof of concept of the algorithms in the late 1990s to the mid 2000 (Böckmann et
al., 2005; Müller et al., 1999a; b; Müller et al., 1998; Müller et al., 2003; Müller et al., 2000;
Veselovskii et al., 2002; Veselovskii et al., 2004). The inversion methodologies were improved to the
point that it became possible to derive profile-like microphysical data products (Kolgotin and Müller,
2008; Müller et al., 2011; Veselovskii et al., 2009) In recent years significant efforts have been made in
automating the methodology developed by Müller et al. (1999a) to the point that unsupervised, real-
time data analysis has become possible (Müller et al., 2014).

Despite this encouraging progress we started following another path of data analysis, i.e. the use of
Artificial Neural Networks also known as Neural Networks (NN) in order to explore if NN or a
combination of NN with the traditional mathematical algorithms could in future increase data
processing speed and the quality (accuracy and precision) of the data products which in part are still not
meeting the requirements requested by the climate modeling community. We also want to explore what
types of combinations of backscatter and extinction coefficients (in terms of number of channels) could
potentially deliver some of the data products we are interested in.

In this contribution we present for the first time results of a large scale sensitivity analysis in which
we tested various simple NN configurations and developed a successful NN design during a 5-years
effort to explore if NN can be used for support in data inversion. NNs are inspired by biological neural
networks (biological neurons) of our human brain consisting of $10^9$-$10^{10}$ neurons (small information
processing elements) which communicate through an interconnected network (approximately $10^4$
connections per element). NN works as massively parallel distributed computing networks, and are
similar to biological neural systems in their main characteristics. Generally NN models are similar to
well-known statistical models, e.g. non-linear regression; however the nomenclature of NN is unlike
compared to that in statistics. In order to cross-validate an NN model, selected datasets are divided into



training set and test set where the independent variables are termed as input and NN estimated values are called output.

We had to start with literally no experience in this research area, in part in view of lack of appropriate literature with regard to aerosol lidar applications. We tested the most traditional NN which is the feed forward backward propagation method. We kept the input data comparably simple but still

complex enough so that we could simulate atmospheric aerosol conditions to a reasonable degree. We tested various combinations of data that can be obtained from high-end multiwavelength lidar and less developed lidar in order to test if there is a threshold to the necessary number of data that could make NN useful for our purposes. It is clear that such a study is not only complex but time consuming as NNs exist in a variety of designs. There also is the question of how representative the input data have to be

for the training, testing, and validation steps. For example we could not yet investigate the impact of measurement errors of the optical data on the performance of the NN that we used in our study. Nonetheless this study provides the first insight to the potential usefulness of neural networks for the analysis of lidar data and the study points toward directions of future research work in that area.

Figure 1 shows a flow chart of the overall procedures of our ANN study for the analysis of

multiwavelength Raman/HSRL lidar data. Section 2 will provide an overview on our NN design. We will describe the choice of our training data and the design of our simulation study. We tested more than fifty different network topologies, among them the most important twenty five network topologies are summarized in this study. The best NN topology which is feasible for our work was selected from those twenty five topologies. See the appendix section for details of network topologies. Section 3 will

describe the results of our main target of our sensitivity study, i.e. retrieving effective radius and, as a byproduct the complex refractive index for various configurations of input data. We focus on some key data combinations which are important of stand-alone multiwavelength lidar and future space-borne lidar missions in which a down-scaled multiwavelength lidar might be coupled with passive remote sensing instruments. Section 4 will present a summary and outlook.




## 2   Methodology

### 2.1   Feedforward Backpropagation Multilayer Perceptron Neural Network

Figure 2 shows a schematic display of the general feedforward multilayer perceptron neural network (FFBP-MLP). In its most simple design a FFBP-MLP consists of three layers, i.e., the input

layer, the hidden layer, and the output layer. In the FFBP-MLP architecture a functional link is interconnecting these layers (Hagan et al., 1996). The neurons are linear or nonlinear computing elements. Adjacent layers are connected to each other by neurons, but there are no lateral connections between neurons within one layer, however feedback connections are possible.

The input variables which can be denoted as $I = [i_1, i_2 \ldots i_n]$, are functionally linked with the main

processing elements, i.e., the neurons. Each single neuron is given a relative weight $W = [w_1, w_2 \ldots w_n]$ which determines the impact of each input. These network weights are adaptive coefficients within the network. The weights determine the proper intensity of the input signal by applying a bias value $b_k$, which is a random initial value i.e., real number multiplied with a weight value. The neuronal outputs are generated by the summation block which adds up all of the weighted input signals algebraically. The

information in this network flows in one direction, from input layer to output layer, via the hidden layer. The forward propagation step starts with uploading an input pattern into the input layer. In our study this means that optical data sets are presented to the network. The network then processes the data in the hidden layer and converts the calculated signals of the hidden layer to the output layer.

We trained our neural network model with an algorithm, i.e., perceptron named the *generalized*

*delta rule*. This delta rule processes derivatives by a simple chain rule called backpropagation (Werbos, 1994) in which the network errors i.e., the difference between true solutions and neural network generated solutions propagate backward and again check for new weight values. During this weight correction procedure, the configuration of the optimum values between output values (e.g., effective radius, real and imaginary parts of refractive index) and target values (their true solutions) are chosen by

computing their differences. In this study, training means a set of input parameters is used in a





combined fashion to search for a specific targeted parameter by repeated adjustments of weights and a fixed random bias value, on the basis of a comparison between the target and network-generated output, until the output ends up in an optimal correlation with the target. The hidden layers use a nonlinear activation function (Hagan et al., 1996) and the whole MLP model becomes truly nonlinear. The

purpose of using a nonlinear activation function is that it introduces nonlinearity into the neural network to solve our linearly inseparable retrieval problem. The activation function for each neuron is the sum of all its input values multiplied by their corresponding connection weights. Once the activation neuron is computed, the output values $O_k$, as shown in figure 2 can be easily determined by applying a transfer function which transforms the output signals into NN estimated target values.


## 2.2 Workflow of the FFBP- MLP Neural Networks Model

Figure 3 shows a simple graphical outline of the FFBP-MLP neural network model that we used for the training phase. The Neural Network Toolbox in MATLAB software (version 2012b) has been used to design the NNs we used in our study. We limited our study to five basic combinations of backscatter

and extinction data. Figure 3 shows the approach that we used when effective radius is chosen as target parameter (output neuron). 5 hidden neurons are used in a single hidden layer. The minimum and maximum values of the input and target parameters (input and output neurons) are shown in the red colored boxes. When we inserted the input values (input parameters/neurons) in the model, no values of output were used as input, which makes the model very robust to construct an input - output mapping

relationship. The complex refractive indices (real and imaginary part), though not the main target of our study, were also trained by using the above mentioned model (not shown here).

There exists a variety of network designs and types of neural networks, each of them can be suited to particular applications. We considered a number of network architectures and parameters in order to determine the optimum model configuration of our study. Among the configurations we tested the

feedforward backpropagation multilayer perceptron performed the best training results on the basis of the *coefficient of determination* ($R^2$) between the known (true) value and the value obtained from our



NN. The coefficient of determination statistic shows the normalized information of the goodness of fit of a model. It describes how strongly the regression straight-line estimates the true data points and provides mean squared errors of training, testing, and validation data. We find five hidden neurons in a single hidden layer to be the best choice for our research work. This decision was based on the fact that we obtained the minimum of the Mean Squared Error (MSE) in the training phases in which we used different numbers of hidden neurons.

We used one FFBP-MLP model for all training and simulation cases. Figure 4 shows the workflow of how the optical data were used for training the NN. At the beginning we insert the input and output parameters in the neural network. Then the network activation function is used, followed by the allocation of transfer functions and number of hidden neurons.

We randomly selected a subset of 70% of the input data for training our model. We used 20% of the data for testing the performance of the model. The remaining 10% of the data were used for the validation step. The validation step allows us to find the predictive error of the model. A maximum of 1000 epochs was chosen for all training sessions. Epochs means iterations. In each iteration step the model creates a relation between input and output by using the hidden neurons, associated weight values, and hidden layer functions. In NN model the training procedure requires iteratively detecting the perfect weights and biases so that the network errors are minimized via a standard numerical optimization algorithm that optimizes the mean squared error performance function (Hagan et al., 1996). The greater the amount of hidden neurons in the hidden layer, the more unmanageable it becomes to obtain the global optimum (Anctil and Lauzon, 2004). We find that 5 hidden neurons provide a reasonable compromise between complexity of the NN, the work effort to train the network, and to obtain useful insight into NN performance for future studies.

The number of input parameters means the number of neurons in the input layer. We used 3-5 input parameters in this study. The output layer consisted of one neuron, as we were looking for solutions to effective radius, the real part, and the imaginary part of the complex refractive index, separately. We used the hidden layer network functions *Tan-sigmoid* and *Pure-linear* (Haykin, 2007). The network



learning algorithm (training function) was *Trainlm*, Levenberg-Marquardt algorithm (Hagan and Menhaj, 1994). The hidden layer activation function (tan-sigmoid) can be written as

$$f(x) = \frac{1}{1 + e^{-x_i}} \tag{1}$$

The parameter $x_i$ is the $i$-th input in the model.

As we set the performance function to mean squared error - MSE, the network will calculate the squared errors based on the difference of the true and estimated values of each output parameter. In our study we take the lowest value of MSE as final results and thus were saved from the trained model. We use three network input-output functions in the neural network toolbox in Matlab as follows: remove

constant rows (RCR), standard deviation function (*mapstd*), and *mapminmax* function (Hagan et al., 1996). The use of the RCR function allows us to remove any possible rows with constant values in order to ensure the statistical robustness of this study, and to keep the maximum number of dissimilar data samples. We often need to use a variable scaling of input–output values to achieve the best results. The network uses the *mapminmax* function for pre-processing and post-processing of the data in a way

such that the training phase data are scaled between -1 to +1. This means all variables fall between the values -1 and +1. After that step we normalize the mean and standard deviation of the training dataset with the *mapstd* function so that the network's input values and target values transform to zero mean and unity standard deviation.

## 2.3 Format of Retrieval Results and Errors


In the results section we will show correlation plots along with some statistics, i.e., reduced chi-square test values, Pearson's correlation coefficient values (r), and adjusted R-square values ($R^2$). Based on the correlation coefficient values of all investigated parameters in this study, a summary figure is shown at the end of results section. For estimating the errors in the simulations and the assessment of

the models that we investigated, the reduced chi-square ($X^2$) test was used. The rationale for using these



different methods is as follows. We have used linear fitting and reduced $X^2$ test which are good choices to predict the goodness of fit of our retrievals against their true solutions.

Pearson's correlation coefficient values (r) are used to see the degree of linear association between the original and simulated values of the investigated parameters. From the two key properties of Pearson's r, which are magnitude and direction, we can easily understand the state of relationships that exists between two variables.

Adjusted $R^2$ also suggests the goodness of fit of the linear regression results. Here, we are using a linear model to fit our simulated data against the original values and the main use of $X^2$ is to test the goodness of the regression lines, and identifying the slope and intercept with respect to our data. As we have two regression parameters (slope and intercept), the number of the degrees of freedom (DOF) minus two is used in the computations. We calculated the residuals as

$$X^2 = \left(\frac{1}{DOF}\right) \times \sum R_i \qquad (2)$$

where $R_i$ = Original values – Simulated values.

The retrieval errors (squared relative error) for effective radius and complex refractive index are calculated as

$$Squared\ relative\ error, \delta x^2 = \sqrt{\frac{1}{N} \sum_{i_{min}}^{i_{max}} \frac{(x_0 - x)^2}{x^2}} \times 100\ (\%) \qquad (3)$$

where N = number of data points used for simulation, $x_0$ describes the NN estimated values and $x$ describes the true values, errors are shown as percentage.

## 2.4 Input Datasets for the FFBP- MLP

In our study, the extinction and backscatter coefficients at the wavelengths 355, 532, and 1064 nm were generated from monomodal logarithmic-normal aerosol particle size distributions using a Mie scattering algorithm (Bohren and Huffman, 1998). Table 1 shows the values of particle mean radius,





geometric standard deviation (mode width), complex refractive index (real and imaginary part), and particle effective radius that we used for the computations of the optical input data. The numbers in

table 1 cover a realistic range of atmospheric particle size distributions. Table 2 summarizes the combinations of optical data that we used for our ANNs. The goal was to find out how well the different optical data combinations allow us to derive effective radius, and the real and imaginary part of the complex refractive index.

The common FFBP-MLP neural network was applied to five basic input combinations. We used

three different size ranges of effective radii for which we tested our NN. We had to do this in order to keep the computation time to a reasonable limit given the computer resources we had at hand. Combining all data between 10 to 500 nm results in 817742 individual data for the training phase for which we need a computation time of more than 3 hours at least. Data downscaling is applied to reduce the computation time. It also allows us, as a by-product, to investigate in a first step the effectiveness of

our NN to identify the properties of the particles in the ultrafine mode, fine mode (accumulation mode), and coarse mode. The coarse mode was not fully covered in this study and a more refined separation into the three particle modes will be done in a future study.

For the retrieval of $r_{\mathrm{eff}}$ in the range from $10 - 100$ nm we used 165240 individual data points for the training step and 165240 dissimilar data for the NN simulation, results and discussions will be shown in

our future study. 250181 different data points were used for the training, and 250181 separate data points were used for the simulations in the $110 - 250$ nm mean-radius range. 402321 different data points were used for the training and 402320 separate data points were used in the simulation in the $260 - 500$ nm mean-radius range. The values of the imaginary part were limited to $0 - 0.1$. We used 453255 individual data points for the training and 453100 different data points for the simulations in the $10 -$

500 nm mean-radius range.

Figure 5 describes how we prepared the optical data for our tests. Before processing the input parameters with the model we applied a Fisher-Yates shuffle algorithm (Paul, 1948) to all data. We had to shuffle the database to avoid any possible bias of data selection during the training phase and the



blind-test phase. This algorithm randomly permutes N elements by exchanging each element with a

random element from *i* to N. All data are inserted in a matrix format in Matlab (www.mathworks.com) and shuffling is done by the Matlab *randperm* (random permutation) function. This function randomly shuffles the whole dataset on a row-by-row basis, i.e., the sampling is done with replacement. The intrinsic structure of the data remains unchanged after the process of random shuffling. The individual values are randomly distributed and we can choose the model training and blind-test data portions

without creating a bias of any particular data type or distribution. After shuffling, 50% of the data were selected for the training part and the remaining 50% of the data were chosen for carrying out the blind-test. In our study, the blind-test means that the data that were not used in the training phase of the NN but were used to test the performance of our algorithm in retrieving the parameters of interest, i.e., the output. Here, the portion of the training data is further subdivided into 'training', 'testing', and

'validation' in proportion of 70%, 20%, and 10%, respectively. Training and blind test datasets are prepared in such a way that they are statistically representative (i.e. mean, standard deviation, median, maximum-minimum values of both datasets are similar) for the whole set of data used in our study. This fifty-fifty data sharing is a good approach for simulating the investigated parameters against the trained data in our model.


## 3 Results and Discussion

The potential of the Artificial Neural Networks in retrieving atmospheric particle parameters (effective radius and the mean complex refractive index) have been implemented in extensive simulations and sensitivity analyses (Mamun, 2014). Sensitivity studies with a number of various input

combinations of backscatter and extinction data were tested to answer the questions mentioned in the introduction section. Our results will be compared with the classical lidar inversion algorithm methods (minimum a priori information used) in our future works.

In this contribution we analyze the main features of our ANN in retrieving particle effective radius. We did not train this ANN with regard to the real and imaginary part in the sense that we tried to



optimize the retrievals for these two parameters using our common FFBP-MLP model. We were mainly interested in the robustness of the chosen ANN (for effective radius retrieval) how well it can also retrieve the complex refractive index. We will in future work develop another ANN that will be specifically optimized with regard to retrieving the complex refractive index.

The ANN method applied in this study did not take into account any a priori information. We
calculated the extinction based Ångström exponent ($å_{355/532nm}$) in the value ranges between -0.5 to < 0, 0 to < 0.5, 0.5 to < 1, 1 to < 1.5, 1.5 to < 2, and 2 to 2.5 for effective radii ranging from $110 - 250$ nm. Additionally $å_{355/532nm}$ was calculated in the ranges between -0.5 to < 0, 0 to < 0.5, 0.5 to < 1, and 1 to 1.5 at $260 - 500$ nm of the corresponding effective radius values. Lidar ratios at 532 nm were computed for several ranges, i.e. < 20 sr, 20 to < 40 sr, 40 to < 60 sr, 60 to < 80 sr, 80 to < 100 sr, and above 100
sr. Again we stress the fact that we were mainly interested in the feasibility of using ANN for analyzing lidar data, and the chosen intervals reflect in a rough sense various size ranges of ultrafine, fine and coarse mode particles (in terms of their Ångström exponents) and the combination of Ångström exponents with lidar ratios reflect different aerosol types (Burton et al., 2012; Burton et al., 2014), e.g., values of $40 - 60$ sr at 532 nm can be regarded as moderately absorbing urban haze (if the Ångström
exponent is above 1), whereas values of $60 - 80$ sr or above may be representative of highly-light absorbing pollution. The results of lidar ratios at 355 nm will be included in our future works.

Figures $6 - 9$ show the results of the simulated values of the investigated parameter versus their true values for various ranges of extinction Ångström exponents. Figures $10 - 13$ show correlation plots of the true versus the simulated parameters based on the various lidar ratios (based at 532 nm). Tables $3 -$
6 show the statistical analysis of the various training and simulation sessions for the three ranges ($110 - 250$ nm, $260 - 500$ nm, and $10 - 500$ nm) of mean radius considered in our study.

We tested different combinations of backscatter and extinction coefficients. As explained in the introduction, the combination of 3 backscatter and 2 extinction coefficients is important as this combination is used in the currently most evolved multiwavelength Raman lidars, e.g.,
(http://www.earlinet.org/) and high-spectral-resolution lidars (Müller et al., 2014). We tested lidar data



combinations that will be important in the context of the synergy of lidar with passive remote sensors with the latter providing optical data at other wavelengths than the ones tested for the lidar set-ups. Our study uses the combination of backscatter and extinction data, which can be seen from the four different combinations used in this work consisting of at least 1α+1β in all cases. In addition we use one case
consisting of only backscatter data (3β case).

## 3.1 Retrieval of Particle Effective Radius

When retrieving atmospheric particle parameters, generally a combination of backscatter and extinction coefficients at various wavelengths is essential for the stability and consistency of the
inversion algorithm methods, i.e., the combination of at least 3β (for aerosol lidar purposes the wavelengths are 355, 532, and 1064 nm) and 2α data (wavelengths are at 355 and 532 nm) is necessary to estimate with acceptable accuracy of the particle parameters investigated in our study (Müller et al., 1999a; b; 2000; 2001). Moreover, adding more data channels provides better results if the chosen wavelengths are outside the interval 355 − 1064 nm, as in general the information content of optical
data in those wavelength ranges below 355 and above 1064 nm may contribute significantly to the information content in the wavelength band between 355 and 1064 nm. We will investigate this effect in future studies as it will be also important in the context of combing lidar with passive remote sensing data.

Our ANN analysis suggests that it is important to have at least a combination of 1α (355 or 532 nm)
and 2β (532 nm and 1064 nm) for the retrieval of $r_{\text{eff}}$ in the particle radius interval between 110 − 500 nm. The results of the use of 3β + 2α datasets in our ANN model are in nice agreement with previous studies carried out with inversion algorithms. A combination of only 3β optical data can also retrieve the investigated parameters in all size ranges, however with reduced accuracy compared to the other four data combinations investigated in our study. This result shows the importance of using extinction
data. The results we obtain from our ANN model indicate the loss of retrieval accuracy if extinction information is omitted and confirms previous studies that pointed out to the fact acceptable results for



microphysical parameters cannot be obtained if only backscatter data are used; we stress the fact that the retrieval situation likely will become worse as soon as measurement errors are considered.

The importance of using backscatter coefficients was exemplified in one study (Böckmann et al., 2005) where the degree of ill-posedness due to backscatter data was found to be less compared to extinction data, however the extinction profiles in combination with backscatter profiles are very important for the best retrieval accuracies, as we find from our NN simulations. The simulation results for $r_{eff}$ of aerosol mean radii between $110 - 250$ nm and $260$ nm - $500$ nm, see tables 3 and 4, show moderately good Pearson's r and adjusted $R^2$ values for all ranges of extinction Ångström exponents when a combination of $3\beta+2\alpha$ is provided as input in the model. Additionally the MSE values are very low for all size ranges. This result suggests that the network's performance quality is very high.

The multiple correlation plots are shown in figures 6 - 7. The best results are achieved for the $3\beta+2\alpha$ data combinations and both size ranges, as can be seen from simulation A. The simulations B, C, and, D show no significant differences in the retrieval quality of the parameters. Simulations with the combination of $1\alpha$ (355 or 532) with $3\beta$, and $2\beta$ (532, 1064) with $1\alpha$ at 532 nm show similar retrieval quality regarding effective radius.

In the radius range of $110 - 250$ nm the correlation plots (Figure 10) based on various lidar ratio ranges suggest that values < 20 sr and values from 20 sr to < 40 sr have the highest correlations; values below 20 sr usually are not measured for aerosol particles, however we still tested this lidar ratio range below 20 sr in order to check the robustness of our ANN scheme. The correlation tends to slightly decrease with increasing lidar ratio. In the case of $3\beta$ (simulation E), significantly stronger correlation is found when the lidar ratio is below 20 sr. However for lidar ratios above 20 sr the association is found to be moderate to strong. Almost the same trend is seen in the particle radius range from $260 - 500$ nm (figure 11) where the highest correlation is found for lidar ratios < 20 sr. In our model, $3\beta$ (the absence of $\alpha$) clearly shows the importance of using extinction profiles in retrieving $r_{eff}$ (over 100 nm mean radii) with acceptable accuracy. We speculate that one of the possible reasons for such strong correlations which have been revealed in our study for the first time is the pre-allocations of various





values of extinction based Ångström exponents before the allocation to lidar ratio is carried out by the ANN. Plausibly, ANNs as an intelligent data mining technique, is capable of finding out this intricate

association among Ångström exponent, lidar ratio, and particle parameters, details of which has been shown for the first time in this study.

## 3.2 Retrieval of Real and Imaginary Part of the Refractive index

The simulation results of the refractive indices show similar trends for the real and imaginary parts.

However, retrievals of the real parts are more accurate compared to the imaginary part. In both cases $3\beta+2\alpha$ data combinations show the best results followed by the other data combinations as follows:

Imaginary part: $3\beta+2\alpha > 3\beta > \alpha$ at $532+3\beta > \alpha$ at $355+3\beta > \alpha$ at $532+2\beta$ (532, 1064).

Real part: $3\beta+2\alpha > \alpha$ at $355+3\beta > \alpha$ at $532+3\beta > \alpha$ at $532+2\beta$ (532, 1064) $> 3\beta$.

Here, for the imaginary part we can see better retrievals from only $3\beta$ over $3\beta+1$ $\alpha$. The

computation of Neural Networks relies upon numerical calculation of input and output parameters and associated weight vectors in the hidden layer, the accuracy of outputs depend on successful selection of bias values and weight vectors. These overall procedures are usually performed by trial and error methods to achieve the optimum results. In this study, for the retrievals of imaginary part (output) from only $3\beta$ coefficients (inputs) can outperform $3\beta+1\alpha$ suggesting a more refined selection of bias and

concomitant weight vectors to achieve optimum imaginary refractive index values.

The simulated imaginary part that is related to the extinction-based Ångström exponent values from -0.5 to $< 0$ has the best correlation (figure 8). The next best correlation is followed by $å_{355/532nm}$-values between 0 to $< 1$. Weak to no correlation is found for $\alpha_{355/532nm}$-values $\geq 1$.

The simulations for the real part (figure 9) show a trend that is similar to the one we find for the

imaginary part, though there is a slightly better accuracy. Very strong correlation is found for the $3\beta$ case with respect to the retrieval of the imaginary part and strong to moderate correlation is found for the real part.





The figures (12 − 13) that show the simulations in dependence on the lidar ratio show different trends for the imaginary and real parts. In general, the lidar ratio range from 40 sr to < 60 sr has the best
correlation with the imaginary part. The imaginary parts that are related to lidar ratios < 20 sr and to lidar ratios in the interval 20 < 40 sr have a moderate relation, except for the case of $1\alpha+2\beta$. Lidar ratios above 60 sr show moderate to strong correlations. Only the $3\beta$ data combination has moderate to strong correlation for all lidar ratios which is unlike the real-part retrievals where we find weak association for lidar ratios <60 sr. The correlation increases with increasing lidar ratios. Other combinations related to
the real part show moderate to strong correlations for lidar ratios < 100 sr. Lidar ratios ≥ 100 sr show very strong correlations.

If we compare the results of the Ångström-based and lidar ratio-based results (see figures 8 - 13) we see a better accuracy for the real part compared to the imaginary part. Simulations for the refractive index also suggest that we can achieve acceptable accuracy even if we use a reduced number of input
neurons (input data combinations). The results of the $3\beta + 2\alpha$ data combination show good agreement with inversion algorithm methods with increased retrieval accuracy as shown in tables 5 and 6.

The retrieval errors for the real part of refractive index are between ±6.71 to ±10.41% whereas the imaginary parts show retrieval errors between ±31.3% and ±38.3%. This results suggests that the prediction of the real part is more accurate compared to the imaginary part. The situation may be
different if we start training our ANN with specific focus on the real and imaginary part rather than effective radius.

Table 7 and 8 show the statistical information, but split into fine mode (effective radius ≤ 500 nm) and coarse mode (effective radius > 500 nm) particles. We show the results for radii ranges of 110 − 250 nm and 260 − 500 nm. With regard to the mean-radii range of 110 − 250 nm (see table 7) the $3\beta +$
$2\alpha$ combination shows the best output both for fine and for coarse mode particles. The weakest correlation is found for the $3\beta$ data set. All other data combinations reveal comparable strong correlation in the case of the coarse mode.





With regard to the mean-radius range of $260 - 500$ nm (table 8) the simulation show weak to moderate correlations for fine mode particles. The exception is the $3\beta + 1\alpha$ (355) data combination. The

poorest correlation again is found for the $3\beta$ data combination. In contrast, all data combinations show strong correlation in the case of the coarse mode except for the $3\beta$ data combination. Overall, the results of the coarse mode simulations show better performance than the simulation with fine mode particles. We conclude that our ANN can estimate particle effective radii larger than 500 nm (i.e. coarse mode, effective radius from 500 to 4000 nm) with higher accuracies than particle effective radii below 500 nm.

In general reduced retrieval errors in all the simulations reveals the robustness of our common neural network model with regard to finding the microphysical properties of atmospheric particle pollution parameters. The application of neural networks confirms the importance of combining backscatter and extinction coefficients which corroborates findings of previous studies with inversion methods of Müller et al. (Müller et al., 2000; Müller et al., 1998; Müller et al., 1999a, b) and Veselovskii et al.

(2002), and eigenvalue analysis carried out by Veselovskii (Veselovskii et al., 2005). From our analysis, we find satisfactory results for $r_{\text{eff}}$, and the real and imaginary parts of the refractive index in terms of MSE and the correlation of the simulated parameter to their true values. All Pearson's correlation coefficient, r values are shown in figure 14 as a summary of the retrieved performances of all investigated parameters in this study.

In some cases, the ANN-calculated values of the aforementioned parameters seem to be more accurate than the conventional mathematical inversion procedures. However, we must keep in mind the simplifications of our study: no errors of the input data were assumed, the test data set was strongly restricted (Ansmann & Müller, 2005) to monomodal size distributions, and the test space may be too constrained in terms of tested PSDs (size parameters). When the complex nature of the relationships

between input and output parameters is totally unknown, FFBPMLP model with one hidden layer works as universal approximator which learns any input-output relationships from a given amount of data (Anastassiou, 2013; White, 1992; Yang et al., 2013; Zainuddin and Pauline, 2008).



## 4 Summary and Outlook

To the best of our knowledge we tested for the first time ANN models for the analysis of multi-wavelength Raman lidar data with regard to retrieving particle effective radius, and the real and imaginary parts of the complex refractive index. For the first time to our knowledge, retrievals of particle parameters of atmospheric pollution ($r_{eff}$, complex refractive index) from multiwavelength Raman or HSRL lidar data has been achieved under simplified conditions (no measurement errors,

monomodal particle size distributions) by means of a feedforward backpropagation neural network. We optimized the ANN with regard to retrieving the effective radius, i.e., the retrieval of the complex refractive index was done with the same model. The results of our ANN model have shown that this method is capable of modelling the complex relationship between optical and microphysical parameters of atmospheric particles for size distributions that range between 0.01μm to 0.50μm. We also show that

our ANN model can retrieve reasonable results from less input information (i.e. 3β, 1α+2β, and 1α+3β), whereas it is still not clear what the minimum input information for traditional inversion algorithm methods needs to be. Some results on that topic can be found in (Chemyakin et al., 2014). Most importantly our retrievals show good agreement with previously tested methods when the combination of 2α +3β is provided as input in the model for the simulation of all parameters we tested (Böckmann et

al., 2005; Chemyakin et al., 2014; Veselovskii et al., 2002; Müller et al., 2001). We hypothesize that the pre-selections of various values of extinction-based Ångström exponents with regard to effective radius before the allocation to lidar ratio is carried out by the ANN and then uses this information to create the strong correlation between particle effective radius and lidar ratios in all PSDs we investigated.

   In the next phase of our work we want to expand the ANN model from 0.51 to 10 μm particle

radius with regard to the aerosol mean radii. We want to introduce more realistic atmospheric situations with regard to the particle size distributions, i.e., multi-modal size distribution, and in a later stage of our work we want to apply the ANN to data from field experiments. Furthermore, we will investigate in more detail how optical properties, i.e. particle Ångström exponents, particle lidar ratios (extinction-to-backscatter ratios) can be used to estimate not only microphysical properties but also the single



scattering albedo. The separation of aerosol types will be tested. Findings from this research will also

contribute to our existing knowledge of various research areas as for example ocean color correction

schemes. We hope that ANN will allow us to excavate the intricate relationships among various

observed parameters of large multi-dimensional datasets and will refine and extract new insight into the

aerosol impact on climate change.





*Acknowledgements* Authors are thankful to REMOTE SENSING CONSULTANTS LTD. of the UK for the financial support on this novel study as a continuation of the author's (M. Mustafa Mamun) master's thesis research.

**Acronyms used**

ANN – Artificial Neural Networks

SSA – Single Scattering Albedo

FFBP-MLP – Feedforward Backpropagation Multi-Layer Perceptron

LIDAR – Light Detection and Ranging

MAPs – Microphysical Aerosol Properties

TCAP- Two-Column Aerosol Project

PSD – Particle Size Distribution

MWL – Multi-Wavelength Lidar

MSE – Mean Squared Error

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

Table 1: Mean radius, mode width, complex refractive index, and particle effective radius used for the
computations of the optical input data.

| Parameter | | | | Value | | | | |
|---|---|---|---|---|---|---|---|---|
| mean radius | | | | 10 nm − 500 nm , in step size of 10 nm | | | | |
| mode width | | | | 1.4 - 2.5, in step size of 0.1 | | | | |
| real part | | | | 1.2 − 2, in step size of 0.025 | | | | |
| imaginary part | | | | $0i$ - $0.1i$, in step size of $9.99\times10^{-6}$ | | | | |
| $r_{eff}$ | range | <0.1 | 0.1 − <0.5 | 0.5 − <1 | 1 − < 2 | 2 − <3 | 3 − <4 | ≥4 | |
| (μm) | N = | 107406 | 492967 | 483837 | 362883 | 135182 | 50761 | 2447 | |
| Real | range | 1.2 − <1.3 | 1.3 − <1.4 | 1.4 − <1.5 | 1.5 − <1.6 | 1.6 − <1.7 | 1.7 − <1.8 | 1.8 − <1.9 | 1.9 − 2.0 |
| part | N = | 106688 | 133361 | 106688 | 106688 | 106688 | 106630 | 106495 | 133117 |
| Imag. part | range | 0 − <0.001 | 0.001 − <0.005 | 0.005 − <0.01 | 0.01 − <0.025 | 0.025 − <0.05 | 0.05 − <0.075 | 0.075 − 0.1 | |
| | N = | 80871 | 40508 | 40508 | 120966 | 201043 | 201316 | 221143 | |





Table 2: Data combinations of backscatter and extinction coefficients that were used in our simulations. EC

denotes extinction coefficient. BC denotes backscatter coefficient.

| Simulation | Input Combinations |
|---|---|
| A | $2\alpha$ (EC$_{355}$+EC$_{532}$) + $3\beta$ ( BC$_{355}$+BC$_{532}$+ BC$_{1064}$) |
| B | $1\alpha$ (EC$_{355}$) + $3\beta$ ( BC$_{355}$+BC$_{532}$+ BC$_{1064}$) |
| C | $1\alpha$ (EC$_{532}$) + $3\beta$ ( BC$_{355}$+BC$_{532}$+ BC$_{1064}$) |
| D | $1\alpha$ (EC$_{532}$) + $2\beta$ (BC$_{532}$+ BC$_{1064}$) |
| E | $3\beta$ ( BC$_{355}$+BC$_{532}$+ BC$_{1064}$) |

Table 3: Summary of effective radius ($r_{eff}$) simulation results (best results) for various input combinations

of optical coefficients in the mean radii range between 110 nm to 250 nm. From Table 3 to table 6 network

performances are depicted as mean squared error (MSE).

| Input combination | Training statistics | | | | Simulation statistics | | | |
|---|---|---|---|---|---|---|---|---|
| | MSE | $R^2$ value Training | $R^2$ value Testing | $R^2$ value Validation | Red Chi Sq ($\chi^2$) | Adjusted $R^2$ | Pearson's r | Retrieval Error (in %) |
| **A.** $3\beta$ (BC$_{355}$, BC$_{532}$, BC$_{1064}$) + $2\alpha$ (EC$_{355}$, EC$_{532}$) | 0.0162 | 0.95376 | 0.95353 | 0.95486 | 0.01478 | 0.90979 | 0.95383 | 16.1 |
| **B.** $3\beta$ (BC$_{355}$, BC$_{532}$, BC$_{1064}$) + $1\alpha$ (EC$_{355}$) | 0.0250 | 0.92777 | 0.9281 | 0.92734 | 0.02156 | 0.86073 | 0.92776 | 20 |
| **C.** $3\beta$ (BC$_{355}$, BC$_{532}$, BC$_{1064}$) + $1\alpha$ (EC$_{532}$) | 0.0256 | 0.92611 | 0.9281 | 0.92569 | 0.02192 | 0.85806 | 0.92631 | 20.2 |
| **D.** $2\beta$ (BC$_{532}$, BC$_{1064}$) + $1\alpha$ (EC$_{532}$) | 0.0272 | 0.92123 | 0.92137 | 0.92148 | 0.02299 | 0.84923 | 0.92154 | 21 |
| **E.** $3\beta$ (BC$_{355}$, BC$_{532}$, BC$_{1064}$) | 0.0749 | 0.76303 | 0.7624 | 0.7619 | 0.04384 | 0.58258 | 0.76327 | 35.1 |



Table 4: Summary of effective radius ($r_{eff}$) simulations results (best results) for various input combinations of optical coefficients in the mean radii range between 260 nm to 500 nm.


| Input combination | Training statistics | | | | Simulation statistics | | | |
|---|---|---|---|---|---|---|---|---|
| | MSE | $R^2$ value Training | $R^2$ value Testing | $R^2$ value Validation | Red Chi Sq ($\chi^2$) | Adjusted $R^2$ | Pearson's r | Retrieval Error (in %) |
| **A.** 3β (BC$_{355}$, BC$_{532}$, BC$_{1064}$) + 2α (EC$_{355}$, EC$_{532}$) | 0.0801 | 0.94492 | 0.94529 | 0.94532 | 0.07197 | 0.89238 | 0.94466 | 17.23 |
| **B.** 3β (BC$_{355}$, BC$_{532}$, BC$_{1064}$) + 1α (EC$_{355}$) | 0.0941 | 0.93483 | 0.93539 | 0.93578 | 0.08265 | 0.87378 | 0.93476 | 19.1 |
| **C.** 3β (BC$_{355}$, BC$_{532}$, BC$_{1064}$) + 1α (EC$_{532}$) | 0.0939 | 0.93521 | 0.93453 | 0.93579 | 0.08258 | 0.87405 | 0.9349 | 19.3 |
| **D.** 2β (BC$_{532}$, BC$_{1064}$) + 1α (EC$_{532}$) | 0.0978 | 0.93246 | 0.93272 | 0.93224 | 0.0849 | 0.86889 | 0.93214 | 19.02 |
| **E.** 3β (BC$_{355}$, BC$_{532}$, BC$_{1064}$) | 0.421 | 0.66091 | 0.6613 | 0.66408 | 0.184 | 0.43968 | 0.66309 | 39.2 |

Table 5: Summary of imaginary part of refractive index simulations results for various input combinations of optical coefficients in the mean radii range between 10 nm to 500 nm.

| Input combination | Training statistics | | | | Simulation statistics | | | |
|---|---|---|---|---|---|---|---|---|
| | MSE | $R^2$ value Training | $R^2$ value Testing | $R^2$ value Validation | Red Chi Sq ($\chi^2$) | Adjusted $R^2$ | Pearson's r | Retrieval Error (in %) |
| **A.** 3β (BC$_{355}$, BC$_{532}$, BC$_{1064}$) + 2α (EC$_{355}$, EC$_{532}$) | 3.03 E-04 | 0.83435 | 0.83382 | 0.83461 | 2.04541 E-04 | 0.69925 | 0.83621 | 31.3 |
| **B.** 3β (BC$_{355}$, BC$_{532}$, BC$_{1064}$) + | 3.95 | 0.77703 | 0.77975 | 0.77685 | 2.35567 | 0.60334 | 0.77804 | 35.81 |



| Input combination | MSE | | | | | | |
|---|---|---|---|---|---|---|---|
| 1α (EC₃₅₅) | E-04 | | | | E-04 | | |
| **C.** 3β (BC₃₅₅, BC₅₃₂, BC₁₀₆₄) + 1α (EC₅₃₂) | 3.83 E-04 | 0.78396 | 0.7847 | 0.78544 | 2.31733 E-04 | 0.6159 | 0.78479 | 35.33 |
| **D.** 2β (BC₅₃₂, BC₁₀₆₄) + 1α (EC₅₃₂) | 4.54 E-04 | 0.73885 | 0.74149 | 0.7395 | 2.41063 E-04 | 0.54927 | 0.74113 | 38.27 |
| **E.** 3β (BC₃₅₅, BC₅₃₂, BC₁₀₆₄) | 3.94 E-04 | 0.77835 | 0.77715 | 0.77797 | 2.34645 E-04 | 0.60718 | 0.77922 | 35.73 |


Table 6: Summary of real part of refractive index simulations results for various input combinations of optical coefficients in the mean radii range between 10 nm to 500 nm.

| Input combination | Training statistics | | | | Simulation statistics | | | |
|---|---|---|---|---|---|---|---|---|
| | MSE | $R^2$ value Training | $R^2$ value Testing | $R^2$ value Validation | Red Chi Sq ($\chi^2$) | Adjusted $R^2$ | Pearson's r | Retrieval Error (in %) |
| **A.** 3β (BC₃₅₅, BC₅₃₂, BC₁₀₆₄) + 2α (EC₃₅₅, EC₅₃₂) | 1.18 E-02 | 0.89072 | 0.89246 | 0.8912 | 0.00914 | 0.79549 | 0.8919 | 6.71 |
| **B.** 3β (BC₃₅₅, BC₅₃₂, BC₁₀₆₄) + 1α (EC₃₅₅) | 1.53 E-02 | 0.85547 | 0.8554 | 0.85505 | 0.01081 | 0.73518 | 0.85743 | 7.64 |
| **C.** 3β (BC₃₅₅, BC₅₃₂, BC₁₀₆₄) + 1α (EC₅₃₂) | 1.59 E-02 | 0.84952 | 0.84712 | 0.84909 | 0.1111 | 0.72471 | 0.8513 | 7.80 |
| **D.** 2β (BC₅₃₂, BC₁₀₆₄) + 1α (EC₅₃₂) | 1.77 E-02 | 0.83043 | 0.83068 | 0.8298 | 0.01191 | 0.6909 | 0.8312 | 8.25 |
| **E.** 3β (BC₃₅₅, BC₅₃₂, BC₁₀₆₄) | 2.82 E-02 | 0.7121 | 0.71352 | 0.71182 | 0.01404 | 0.50875 | 0.71327 | 10.41 |





Table 7: Statistical results of fine and coarse mode particles in the range of $110 - 250$ nm mean radii data
set of particle effective radius.

| Data range | | Input combination | Simulation statistics | | | | |
|---|---|---|---|---|---|---|---|
| | | | Red Chi Sq ($\chi^2$) | Adjusted $R^2$ | Pearson's r | Intercept | Slope |
| $110 - 250$ nm | Fine mode (N = 115933) | A.  $3\beta$ ($BC_{355}$, $BC_{532}$, $BC_{1064}$) + $2\alpha$ ($EC_{355}$, $EC_{532}$) | 0.00659 | 0.57812 | 0.76034 | 0.023 | 1.035 |
| | | B.  $3\beta$ ($BC_{355}$, $BC_{532}$, $BC_{1064}$) + $1\alpha$ ($EC_{355}$) | 0.00903 | 0.51293 | 0.71619 | 0.0404 | 1.062 |
| | | C.  $3\beta$ ($BC_{355}$, $BC_{532}$, $BC_{1064}$) + $1\alpha$ ($EC_{532}$) | 0.00923 | 0.49962 | 0.70684 | 0.045 | 1.045 |
| | | D.  $2\beta$  ($BC_{532}$, $BC_{1064}$) + $1\alpha$ ($EC_{532}$) | 0.00881 | 0.51827 | 0.71991 | 0.044 | 1.061 |
| | | E.  $3\beta$ ($BC_{355}$, $BC_{532}$, $BC_{1064}$) | 0.01321 | 0.32355 | 0.56882 | 0.168 | 0.866 |
| | Coarse mode (N = 134248) | A.  $3\beta$ ($BC_{355}$, $BC_{532}$, $BC_{1064}$) + $2\alpha$ ($EC_{355}$, $EC_{532}$) | 0.02174 | 0.84128 | 0.91721 | 0.056 | 0.913 |
| | | B.  $3\beta$ ($BC_{355}$, $BC_{532}$, $BC_{1064}$) + $1\alpha$ ($EC_{355}$) | 0.03172 | 0.77715 | 0.88156 | 0.052 | 0.894 |
| | | C.  $3\beta$ ($BC_{355}$, $BC_{532}$, $BC_{1064}$) + $1\alpha$ ($EC_{532}$) | 0.03234 | 0.77107 | 0.8781 | 0.0585 | 0.887 |
| | | D.  $2\beta$  ($BC_{532}$, $BC_{1064}$) + $1\alpha$ ($EC_{532}$) | 0.03462 | 0.75435 | 0.86854 | 0.0658 | 0.877 |
| | | E.  $3\beta$ ($BC_{355}$, $BC_{532}$, $BC_{1064}$) | 0.06679 | 0.32716 | 0.57198 | 0.3924 | 0.485 |





Table 8: Statistical results of fine and coarse mode particles in the range of 260 – 500 nm mean radii data
set of particle effective radius.

| Data range | | Input combination | Simulation statistics | | | | |
|---|---|---|---|---|---|---|---|
| | | | Red Chi Sq ($\chi^2$) | Adjusted $R^2$ | Pearson's r | Intercept | Slope |
| 260 – 500 nm | Fine mode (N = 31704) | A.  3β (BC$_{355}$, BC$_{532}$, BC$_{1064}$) + 2α (EC$_{355}$, EC$_{532}$) | 0.00557 | 0.38139 | 0.61757 | -0.1212 | 1.353 |
| | | B.  3β (BC$_{355}$, BC$_{532}$, BC$_{1064}$) + 1α (EC$_{355}$) | 0.00283 | 0.54052 | 0.7352 | -0.1016 | 1.33 |
| | | C.  3β (BC$_{355}$, BC$_{532}$, BC$_{1064}$) + 1α (EC$_{532}$) | 0.00716 | 0.34124 | 0.58416 | -0.1452 | 1.405 |
| | | D.  2β (BC$_{532}$, BC$_{1064}$) + 1α (EC$_{532}$) | 0.00341 | 0.40393 | 0.63556 | 0.0176 | 1.111 |
| | | E.  3β (BC$_{355}$, BC$_{532}$, BC$_{1064}$) | 0.08956 | 0.01675 | 0.12944 | 0.44447 | 0.901 |
| | Coarse mode (N = 370616) | A.  3β (BC$_{355}$, BC$_{532}$, BC$_{1064}$) + 2α (EC$_{355}$, EC$_{532}$) | 0.07707 | 0.88028 | 0.93823 | 0.1701 | 0.885 |
| | | B.  3β (BC$_{355}$, BC$_{532}$, BC$_{1064}$) + 1α (EC$_{355}$) | 0.08878 | 0.85924 | 0.92695 | 0.1974 | 0.866 |
| | | C.  3β (BC$_{355}$, BC$_{532}$, BC$_{1064}$) + 1α (EC$_{532}$) | 0.08813 | 0.85993 | 0.92733 | 0.1997 | 0.864 |
| | | D.  2β (BC$_{532}$, BC$_{1064}$) + 1α (EC$_{532}$) | 0.0914 | 0.85406 | 0.92415 | 0.2071 | 0.859 |
| | | E.  3β (BC$_{355}$, BC$_{532}$, BC$_{1064}$) | 0.18466 | 0.40738 | 0.63827 | 0.8307 | 0.419 |




**Retrieval of Intensive Aerosol Microphysical Parameters
from Multiwavelength Raman /HSRL Lidar:
Feasibility Study with Artificial Neural Networks**

**Databank**
Synthetic optical coefficient data
computed with Mie-scattering algorithm
(See table 1)

**Artificial Neural Networks**
Input Neurons
Hidden layer Neurons
Output Neurons (See figures 2 - 5)

**ANN topology selection**
Best network topology for lidar data analysis:
Feedforward Backpropagation Multilayer Perceptron
Neural Networks (See Appendix section)

**Retrieval accuracy of parameters**

1. Particle Effective Radius ($r_{eff}$) (See tables 3 - 4)

| range | accuracy |
|---|---|
| 110 - 250 nm | $\pm 16$ to $\pm 35$ % |
| 260 - 500 nm | $\pm 17$ to $\pm 39$ % |

2. Imaginary part of complex refractive index
(values between (0 - 0.1$i$) (See table 5)

| range | accuracy |
|---|---|
| 10 - 500 nm | $\pm 31$ to $\pm 38$ % |

3. Real part of complex refractive index
(values between (0 - 0.1$i$) (See table 6)

| range | accuracy |
|---|---|
| 10 - 500 nm | $\pm 7$ to $\pm 10$ % |

See table 7 - 8 for retrieval
statistics of fine and coarse
mode particle effective radius

**Five basic data combinations**
A. 3β+2α
B. 3β+1α (355 nm)
C. 3β+1α (532 nm)
D. 2β (532, 1064 nm)+1α (532 nm)
E. 3β
(See table 2)

**Investigated parameters**
1. Particle Effective Radius ($r_{eff}$)
from 110 - 250 and 260 - 500 nm
2. Imaginary part of complex refractive index
(values between (0 - 0.1$i$), from 10 - 500 nm
3. Real part of complex refractive index
(values between (0 - 0.1$i$), from 10 - 500 nm

| $r_{eff}$ 110 - 250 nm | | $r_{eff}$ 260 - 500 nm | | Imaginary part of CRI (values between (0 - 0.1$i$) 10 - 500 nm | | Real part of CRI (values between (0 - 0.1$i$) 10 - 500 nm | |
|---|---|---|---|---|---|---|---|
| Ångström exponent $\mathring{a}_{355/532nm}$ | Lidar ratio at 532 nm | Ångström exponent $\mathring{a}_{355/532nm}$ | Lidar ratio at 532 nm | Ångström exponent $\mathring{a}_{355/532nm}$ | Lidar ratio at 532 nm | Ångström exponent $\mathring{a}_{355/532nm}$ | Lidar ratio at 532 nm |
| -0.5 to <0 | <20 sr | -0.5 to <0 | <20 sr | -0.5 to < 0 | <20 sr | -0.5 to < 0 | <20 sr |
| 0 to <0.5 | 20 to <40 sr | 0 to <0.5 | 20 to <40 sr | -0 to < 1 | 20 to <40 sr | -0 to < 1 | 20 to <40 sr |
| 0.5 to <1 | 40 to <60 sr | 0.5 to <1 | 40 to <60 sr | 1 to < 2 | 40 to <60 sr | 1 to < 2 | 40 to <60 sr |
| 1 to <1.5 | 60 to <80 sr | 1 to <1.5 | 60 to <80 sr | 2 to < 3 | 60 to <80 sr | 2 to < 3 | 60 to <80 sr |
| 1.5 to <2 | 80 to <100 sr | | 80 to <100 sr | 3 to < 4.5 | 80 to <100 sr | 3 to < 4.5 | 80 to <100 sr |
| 2 to <2.5 | ≥100 sr | | ≥100 sr | | ≥100 sr | | ≥100 sr |
| (See figure 6) | (See figure 7) | (See figure 8) | (See figure 9) | (See figure 10) | (See figure 11) | (See figure 12) | (See figure 13) |

See all figure's summary in figure 14

Figure 1: A flow chart on the overall steps of the feasibility study with ANN for the analysis of
multiwavelength Raman/HSRL lidar data.





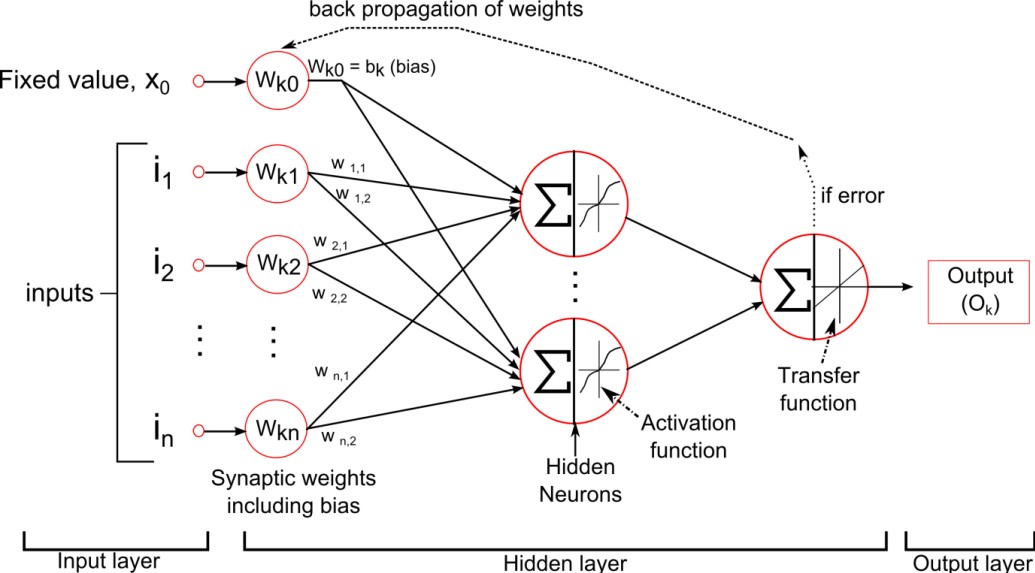

**Schematic of a feedforward backpropagation multilayer perceptron neural network**

Figure 2: Schematic representation of the basic feedforward backpropagation multilayer perceptron neural network that we used in this study.







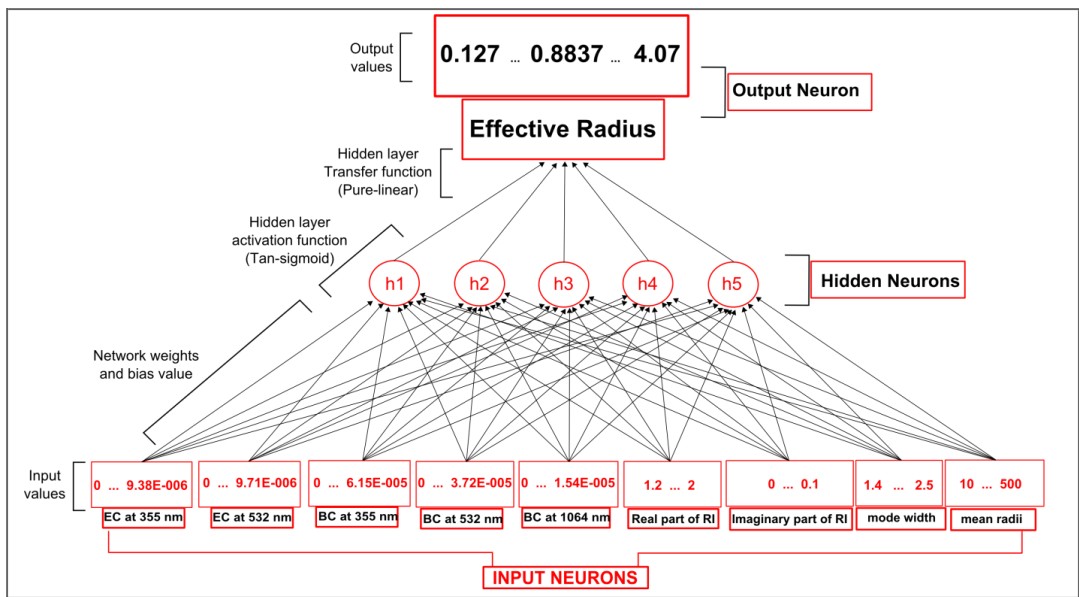

Figure 3: Schematic of our common ANN model for the retrieval of particle effective radius. Results and discussions shown in this study are limited to input neurons of optical coefficients only. Results of complex refractive indices, mode width, and mean radii input neurons are not shown in this study. Numeric values in the red colored boxes show the minimum maximum value of input neurons and output neuron.




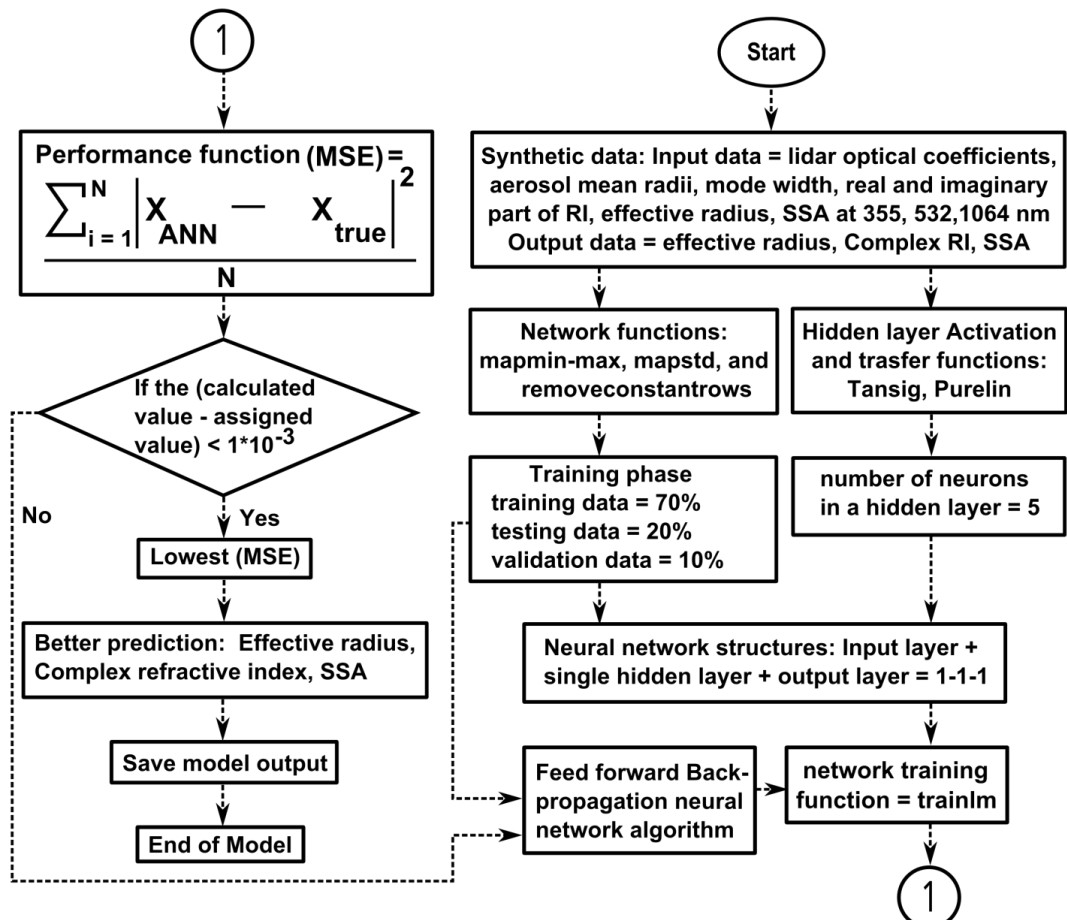

Figure 4: Work flow of the ANN for the analysis of the optical data. We investigated the cases of SSA with
the basic model optimized for particle effective radius, the results and discussions are not shown in this
study and will be provided in future research.



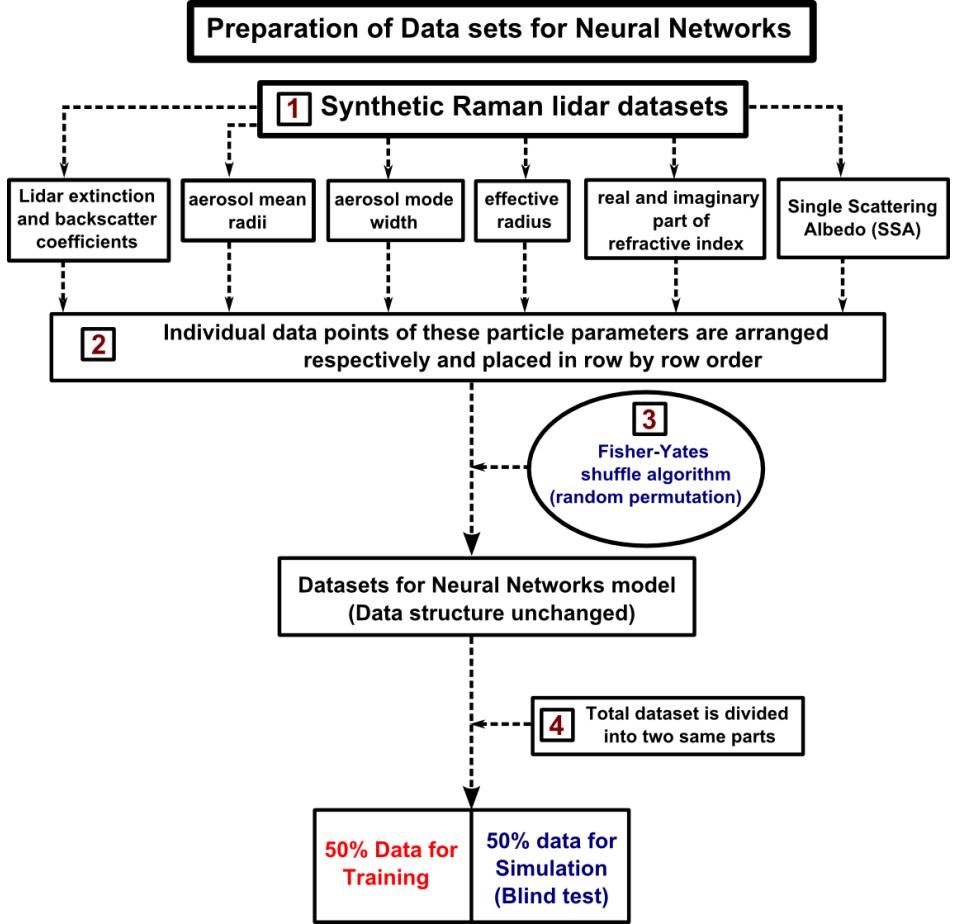

Figure 5: Flow chart depicts the preparation process of our datasets for the neural networks. Results and discussions of SSA are excluded in this study and will be shown in a future contribution.






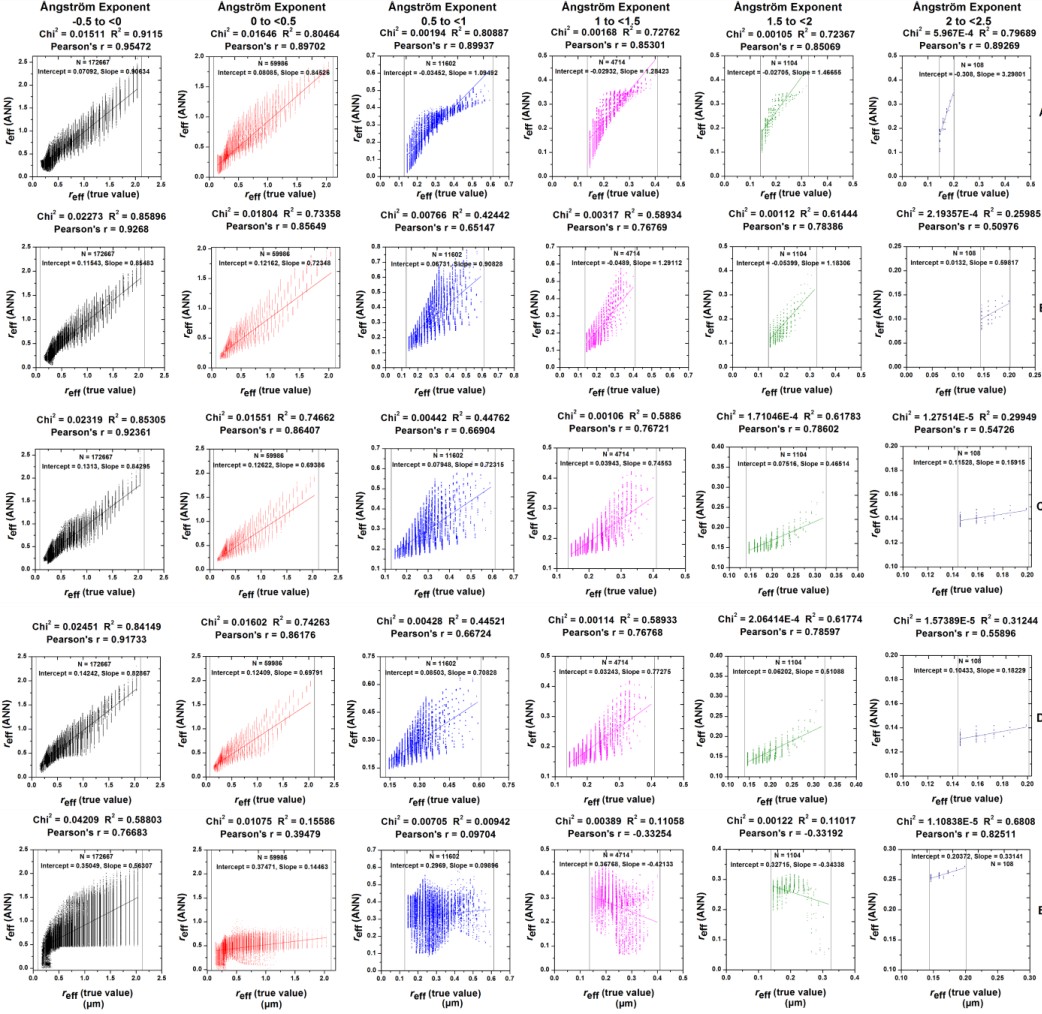

Figure 6: Correlation plots of true vs simulated $r_{\mathrm{eff}}$ for particle radius range from 110 – 250 nm and for different extinction-related Ångström exponents.





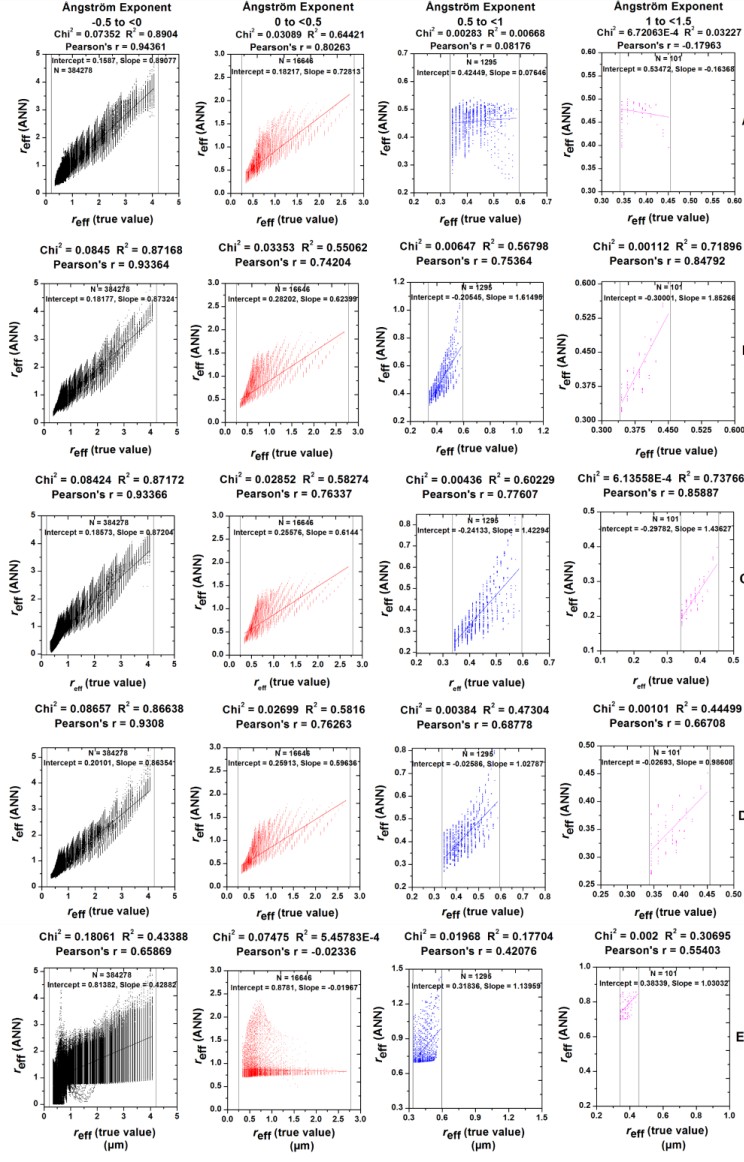

Figure 7: Correlation plots of true vs simulated $r_{eff}$ for particle radius range from 260 – 500 nm and for
different extinction-related Ångström exponents.







Figure 8: Correlation plots of true vs simulated imaginary part for particle radius range from 10 – 500 nm and for different extinction-related Ångström exponents.



Figure 9: Correlation plots of true vs simulated real part for particle radius range from 10 – 500 nm and for different extinction-related Ångström exponents.





Figure 10: Correlation plots of true vs simulated effective radius for particle radius range from 110 – 250 nm for various ranges of lidar ratios at 532 nm.






Figure 11: Correlation plots of true vs simulated effective radius for particle radius range from 260 – 500

nm for various ranges of lidar ratios at 532 nm.







Figure 12: Correlation plots of true vs simulated imaginary part of refractive index for particle radius from 10 – 500 nm (imaginary values between 0 – 0.1) and for various ranges of lidar ratios at 532 nm.






Figure 13: Correlation plots of true vs simulated real part of refractive index for particle radius from 10 – 500 nm (imaginary values between 0 – 0.1) and for various ranges of lidar ratios at 532 nm.



Figure 14: Pearson's correlation coefficient, r value of all investigated parameters shown in this study.





**Appendix**

**Selection of the common FFBP- MLP neural network topology**

In the first step of our work we focus on the most common artificial neural network, i.e. FFBPMLP. We use the Levenberg Marquardt training algorithm (trainlm) which is widely accepted to solving complex input-output problems of large datasets. However, other neural networks were also tested in the process of selecting the best network topology.

The greater the number of hidden neurons the more training time it takes for a successful model run. As we only use one output neuron in our network architecture, we avoided training with a higher number of hidden neurons as this would only increase the training time without achieving any overall better performance. In addition, too many neurons often cause the network to get over-trained or over-fitted. A low number of hidden neurons (i.e. 5, in our model) maintains good generalization abilities of the trained

neural network and prevents overfitting of the output values once the network has been trained (Wilamowski, 2011). A small number of hidden neurons often does not provide the best mean squared error (MSE). However using a low number of neurons allows us to achieve better results when new data patterns are presented to a trained model. A high number of hidden neurons provides better training results but fails to simulate the same results when new data which were not used in the training phase, are

presented to the neural network (Wilamowski, 2003, 2011). In general, trial and error methods are very helpful for deciding how many hidden neurons should be used for a successful multi-layer perceptron (MLP) neural network model (Wilamowski, 2003).

A training-validation-testing data division of 60 -20 -20 (in %) was used for testing ANN #1 – #5 in table 9. We found the best results for ANN#3 which uses five hidden neurons. The MSE values tend to

decrease as the number of hidden neurons increases. The simulation statistics suggest that there is no overall better performance if we use six or seven neurons.

With regard to ANN #6 – #10 we used the data division 70% - 20% - 10%. We found the same patterns with regard to MSE and the hidden neurons. Five hidden neurons were found to be the best choice in both cases of data division, i.e., 60% - 20% - 20% and 70% - 20% - 10%. As we found satisfactory

performance in the case of using five hidden neurons we only used 5 hidden neurons in the network topological runs ANN #11 – #15. We used other types of data division, but no better simulation output was achieved. We achieved the best results in ANN #8 and hence the properties of ANN #8 were selected as a common neural network model for our study.





Table 10 shows the results of a single data division (70-20-10) in $110 - 250$ nm mean radii range data.

In this table, we additionally tested the performance of three other network architectures and training algorithms. In the $110 - 250$ nm data set in table 9, at first we started with the best network design, i.e. ANN #8 (see table 9). With regard to ANN #2 – #4 three other neural networks (cascade forward pattern recognition and layer recurrent) were applied in which five hidden neurons were used. None of these attempts showed better performance than what we obtained on the basis of ANN #1.

Afterwards with regard to ANN #1, we checked the effect of other hidden neurons (i.e. 3, 4, and 6) in ANN# 5-7. We found that a layer of five hidden neurons performs better. Three other training algorithms, i.e., the scaled conjugate gradient (trainscg), the resilient backpropagation (trainrp), and the variable learning rate (traingdx) were tested in ANN #8 - #10. We did this test in order to check if these learning algorithms can provide an optimum output for a moderately large data set, i.e. mean radii data between 110

nm and 250 nm. Among all tested combinations we found that ANN #1 provides the best output. Therefore, this topology was finally selected for our basic ANN model for all data sets and investigated output parameters.

Table 9: Summary of different networks evaluated to select the best network topology for our common

ANN model in the range of $10 - 100$ nm mean radii data set of particle effective radius.

| Tested topologies | Network architecture | Training algorithm | Hidden Neurons | Data Division (%) | Training statistics | | | | Simulation statistics | | |
|---|---|---|---|---|---|---|---|---|---|---|---|
| | | | | | MSE | Training ($R^2$) | Validation ($R^2$) | Test ($R^2$) | Adjusted $R^2$ | Pearson's r | Reduced Chi$^2$ ($\chi 2$) |
| ANN # 1 | Feedforward backpropagation | trainlm | 3 | Train 60 val 20 test 20 | 0.00343 | 0.93864 | 0.93831 | 0.93728 | 0.61932 | 0.78697 | 0.01743 |
| ANN # 2 | Feedforward backpropagation | trainlm | 4 | Train 60 val 20 test 20 | 0.00299 | 0.94646 | 0.9461 | 0.94582 | 0.59349 | 0.77038 | 0.06264 |
| ANN # 3 | Feedforward backpropagation | trainlm | 5 | Train 60 val 20 test 20 | 0.00228 | 0.96046 | 0.96006 | 0.96034 | 0.64919 | 0.80572 | 0.06516 |
| ANN # 4 | Feedforward backpropagation | trainlm | 6 | Train 60 val 20 test 20 | 0.00185 | 0.96724 | 0.96764 | 0.96649 | 0.54476 | 0.73808 | 0.06394 |
| ANN # 5 | Feedforward backpropagation | trainlm | 7 | Train 60 val 20 test 20 | 0.00177 | 0.96852 | 0.96925 | 0.96848 | 0.50794 | 0.7127 | 0.00922 |
| ANN # 6 | Feedforward backpropagation | trainlm | 3 | Train 70 val 20 test 10 | 0.00344 | 0.93829 | 0.93865 | 0.93831 | 0.55859 | 0.74739 | 0.01789 |
| ANN # 7 | Feedforward backpropagation | trainlm | 4 | Train 70 val 20 test 10 | 0.00266 | 0.95249 | 0.95264 | 0.95207 | 0.49873 | 0.70621 | 0.03157 |
| **ANN # 8** | **Feedforward** | **trainlm** | **5** | **Train 70 val 20** | **0.00226** | **0.96089** | **0.9609** | **0.9602** | **0.70687** | **0.84076** | **0.04002** |



| | backpropagation | | | test 10 | | | | | | | |
|---|---|---|---|---|---|---|---|---|---|---|---|
| ANN # 9 | Feedforward backpropagation | trainlm | 6 | Train 70 val 20 test 10 | 0.00181 | 0.96784 | 0.9675 | 0.96847 | 0.55323 | 0.7438 | 0.02781 |
| ANN # 10 | Feedforward backpropagation | trainlm | 7 | Train 70 val 20 test 10 | 0.00164 | 0.97103 | 0.97029 | 0.97087 | 0.37145 | 0.60946 | 0.17046 |
| ANN # 11 | Feedforward backpropagation | trainlm | 5 | Train 70 val 15 test 15 | 0.00206 | 0.96341 | 0.96212 | 0.96257 | 0.62452 | 0.79026 | 0.02765 |
| ANN # 12 | Feedforward backpropagation | trainlm | 5 | Train 80 val 10 test 10 | 0.00208 | 0.96314 | 0.96206 | 0.96368 | 0.61825 | 0.78629 | 0.02801 |
| ANN # 13 | Feedforward backpropagation | trainlm | 5 | Train 90 val 10 test 0 | 0.00226 | 0.95984 | 0.96093 | 0.95963 | 0.67626 | 0.82235 | 0.05122 |
| ANN # 14 | Feedforward backpropagation | trainlm | 5 | Train 50 val 30 test 20 | 0.00204 | 0.96392 | 0.96359 | 0.96424 | 0.52052 | 0.72147 | 0.07078 |
| ANN # 15 | Feedforward backpropagation | trainlm | 5 | Train 60 val 30 test 10 | 0.00210 | 0.96286 | 0.96306 | 0.96438 | 0.61816 | 0.78623 | 0.02657 |

Table 10: Summary of different networks evaluated to select the best network topology for our common ANN model in the range of 110 – 250 nm mean radii data set of particle effective radius.


| Tested topologies | Network architecture | Training algorithm | Hidden Neurons | Data Division (%) | Training statistics | | | Simulation statistics | | |
|---|---|---|---|---|---|---|---|---|---|---|
| | | | | | MSE | Training ($R^2$) | Validation ($R^2$) | Test ($R^2$) | Adjusted $R^2$ | Pearson's r | Reduced Chi$^2$ ($\chi$2) |
| **ANN # 1** | **Feedforward backpropagation** | **trainlm** | **5** | **Train 70 val 20 test 10** | **0.0155** | **0.95603** | **0.95676** | **0.95508** | **0.91429** | **0.95618** | **0.01413** |
| ANN # 2 | Cascade forward Network | trainlm | 5 | Train 70 val 20 test 10 | 0.0194 | 0.94438 | 0.94526 | 0.94589 | 0.8926 | 0.94477 | 0.01725 |
| ANN # 3 | Pattern recognition network | trainlm | 5 | Train 70 val 20 test 10 | 0.0157 | 0.95513 | 0.95583 | 0.95506 | 0.66831 | 0.8175 | 1.92725 |
| ANN # 4 | Layer recurrent network | trainlm | 5 | Train 70 val 20 test 10 | 0.0196 | 0.94394 | 0.94284 | 0.94409 | 0.35289 | 0.59405 | 0.12202 |
| ANN # 5 | Feedforward backpropagation | trainlm | 3 | Train 70 val 20 test 10 | 0.0213 | 0.93849 | 0.93934 | 0.93961 | 0.88162 | 0.93895 | 0.01881 |
| ANN # 6 | Feedforward backpropagation | trainlm | 4 | Train 70 val 20 test 10 | 0.0199 | 0.94288 | 0.9428 | 0.94336 | 0.61398 | 0.94296 | 0.01772 |
| ANN # 7 | Feedforward backpropagation | trainlm | 6 | Train 70 val 20 test 10 | 0.0194 | 0.9444 | 0.94498 | 0.94551 | 0.89227 | 0.9446 | 0.0173 |
| ANN # 8 | Feedforward backpropagation | trainscg | 5 | Train 70 val 20 test 10 | 0.0272 | 0.92104 | 0.92048 | 0.92133 | 0.49699 | 0.70498 | 0.1435 |
| ANN # 9 | Feedforward backpropagation | trainrp | 5 | Train 70 val 20 test 10 | 0.0265 | 0.92319 | 0.92407 | 0.92265 | 0.75014 | 0.86611 | 0.03687 |
| ANN # 10 | Feedforward backpropagation | trainlm | 5 | Train 70 val 20 test 10 | 0.0194 | 0.94438 | 0.94526 | 0.94589 | 0.8926 | 0.94477 | 0.01725 |