# Peer review of "Retrieval of Intensive Aerosol Microphysical Parameters from Multiwavelength Raman/HSRL Lidar: Feasibility Study with Artificial Neural Networks"

_Atmospheric Measurement Techniques, 2016_

## Referee Comment (RC1) · Anonymous Referee #3 · 28 Mar 2016

Retrieving aerosol microphysical parameters from lidar data is an active topic of aerosol research. The manuscript of M.M. Mamun and D. Muller explores the feasibility to retrieve such properties using Artificial Neural Networks, which seems to be a promising approach for future research. Unfortunately, the current form of the manuscript has several weaknesses and omissions that should be addressed before being published.

Major comments:

The authors should give better motivation for their approach. They mention that ANN could "increase data processing speed and quality". However, they are not mentioning

other techniques for the retrieval of aerosol properties, like the linear estimator approach of Veselovskii et al. 2012. This method in particular is fast and can provide the same parameters like the ANN, namely aerosol effective radius and refractive index. Given this, what are the current (or future) benefits of the ANN approach?

The authors do not address the effect of measurement uncertainties in the quality of the results. This is a crucial omission in the analysis. In ill-posed problems, such as the one tackled by the authors, a small change in input data could lead to completely different retrieved solution. Stabilizing (regularizing) the solution is one of the main outcomes of original work like Muller 1999a, b etc. The authors should perform at least some sensitivity test to show that their solutions are stable. Given the processing speed of NN, this should be a straightforward task. Without this, all the results presented by the authors are of limited usefulness. If the results are unstable, the authors should explain how this can be improved in the future (e.g. training the network with perturbed data etc.).

The authors give a very detailed analysis of the network performance for different particle radius and angstrom exponents etc (figs 6 to 13), but they lack an overview of the performance. I would like to see first some overview plots and tables describing the overall performance of the network. These could focus on the "easy" case of 3a + 2b dataset (Case A). In the current form, it is very hard for the reader to extract any useful conclusions.

For the 10 – 100nm range, the authors state that "results and discussions will be shown in our future study". This is strange! Why should part of the analysis be omitted? Are the results very interesting or very bad? The authors should present the results and clearly state the problems and opportunities that they present. Otherwise, the conclusion that the ANN method can model particles from 0.01um to 0.50um is not valid.

Minor comments [line number in brackets]:

[93] Explain what are the requirements of climate modeling community. Why is the processing speed of e.g. Muller et al 2014 not sufficient for climate modeling?

[136] Revise this sentence. Feedback connections are against the definition of feed-forward NN!

[211 – 223] This part reads like a training manual for MATLAB functions. The authors should focus on the algorithms and not on the specific implementations. These information should be moved to an Appendix.

[261] Give a brief description of the computer resources you use. This will give context to the reader concerning the computational effort needed for your approach.

[268 – 275] These numbers should be better summarized in one sentence or a small table.

[290] In a previous paragraph you mention that you use 70% for training, 20% for testing, and 10% for validation. Why do you use an extra 50% of the original data for blind-testing? Give supporting evidence/references that fifty-fifty sharing is a good approach.

[281] Give link to Matlab in the first use of the name. Also, give proper reference to the NN toolbox of Matlab.

Technical comments:

[63] Provide a reference for EarthCARE mission.

[208] The learning algorithm is the LM algorithm., Trainlm is just the MATLAB implementation of this algorithm.

[Tables] You define the scenarios (A, B, . . .) in table 2. You can omit the details in all other tables ("Input combination" column).

[Tables] Take care that all values are provided with the proper significant digits.

[Figures] The font is too small, it should be increased to a legible size. Consider summarizing your results in fewer, clearer plots.

---

## Referee Comment (RC2) · Anonymous Referee #2 · 30 Mar 2016

A one-layer neural network (NN) is generated to estimate effective radius from multi-channel Raman lidar observations. The training and validation data are generated from a Mie code for various mono-modal log-normal size distributions. The best choice of input data is evaluated by considering the correlation of the network's outputs with the validation data, finding that the use of three backscatter and two extinction observations is usually optimal.

I reject this paper as I find the description of the methods excessively vague, consider the experiments poorly designed to answer the questions posed, and do not believe the

data justify the conclusions. The paper was particularly disappointing as Prof. Müller has previously produced exemplary research in this extremely interesting and relevant area.

The omission of uncertainty in the inputs, in my opinion, makes the results of Figs. 6–13 extremely unconvincing. This algorithm is presented with ideal inputs — noise-free simulations of perfectly spherical, mono-modially distributed particles — and yet barely produces results of equivalent quality to the studies cited in the conclusion. I struggle to see this algorithm producing useful results when exposed to the real world of noisy data and uneven particles. To be frank, if this data had been used to conclude that the NN technique is unsuitable for estimating effective radius from lidar data, I would have accepted it with minimal comment.

I must ask what is this paper trying to achieve? By evaluating only simulated data, it fails to show the algorithm is practically useful. By not analysing the validation data with any alternative algorithms, it fails to show the NN technique is any improvement on the status quo. The differences in performance shown are insufficient to establish that any one NN is superior. The omission of uncertainties precludes a sensitivity study. The title proclaims this is a feasibility study. The paper certainly shows that you *can* determine effective radius using a NN but it provides no convincing reason *why* or if this is done sufficiently well to be worthwhile. If the authors simply wanted to show it could be done, they could have written a one-page letter.

I liked that the introduction outlined the broad concept of the technique as this paper is the first application of NN to lidar and readers are unlikely to be familiar with it. Figure 2 is particularly well designed. However, the authors go too far in omitting any details of the algorithm used; simply listing MATLAB functions is insufficient. In particular,

- How, exactly, are the components of the refractive index determined? The paper often implies they are retrieved (such as at L307), but Fig. 3 indicates that they are inputs to the neural net. Fig. 4 implies there is some manner of iteration, but

the text constrains that iteration to the creation of the net.

- Further to that point, I am deeply concerned that the effective radius and refractive index are determined separately from the same input data. Why are they not both outputs of the algorithm? Even in the worst case considered, there are three inputs from which three outputs would be theoretically possible. NNs already present a "black box" and the vague manner in which the refractive index is determined presents the possibility that the same data is being used to determine multiple parameters in separate steps, which is a misuse of the available information.

- How many neural networks were used to produce the data presented — one or five? The text does not clarify if a single network was produced (for case A, taking $3\beta$ and $2\alpha$) and then operated using only some of the inputs or if five separate networks were trained for the cases A–E. The text implies the former, such that I wonder why the authors believe the network will work when presented with fewer inputs? Was it trained with various numbers of outputs? It seems entirely unsurprising that case A has superior performance if that's what the network was built to do.

  The extensive comparisons make the most sense if multiple networks were built, but then I would like to know how continuous the networks are? How do the result of the case A network compare to the case E network on the same simulation? Why would we expect different networks using different inputs to produce equivalent data?

- By your earlier description, the NN contains two components — the weights/ biases applied to the inputs and the activation functions of the nodes. It seems as if the weights were determined during network training, but on L398 you say that the weights/biases are determined by trial-and-error. How important is the selection technique? How many such techniques exist?

As to the biases, L157 states they are fixed and random but L198 indicates they are determined within a numerical optimisation. Which is it?

- If each node has a weight and bias, why do you need to map the input data onto a normalised distribution (L219–224)? What are the standard deviations before mapping? Scaling to $[-1, 1]$ I can understand, but standard deviation scaling can be a non-linear transform of the data that I find concerning. Why was it necessary? "To achieve the best results" doesn't tell me what goes wrong if I don't.

- I agree with the referee #3 that there should be a discussion of why the NN technique is being used when others (by one of the authors and others) are available and could have been optimised for computational time.

  One suspects from L93 that it was used simply to see what would happen. While I've no particular problem with that, it would be good to hear a little more justification of why this particular NN was used over all those available. Why use only one layer? The original submission of this paper indicated that, in addition to evaluating the optimal number of measurements, there was a study of the optimal number of hidden neurons. Though there was too much detail then, it's complete removal provides too little.

- The mean squared error (MSE) is normalised by the degrees of freedom (DOF), being dependent on the number of measurements. However, the backscatter and extinction data have different units, different magnitudes, and very different uncertainties. The definition in Eq. (3) attempts to avoid some of these by normalising by the true value, but this still treats uncertainties in backscatter and extinction as equivalent, which is wrong. The denominator of Eq. (3) should be the uncertainty of the input.

- You often misuse the terms uncertainty and error. Error is the difference between your measurement and the 'true' value. Uncertainty is your estimate

of the likely magnitude of the error. Robust definitions can be found at http://www.iso.org/sites/JCGM/GUM-introduction.htm. Thus, Eq. (3) is an error because it is the difference between the estimated and modelled values whilst your study neglected uncertainties in the input data.

- Aspects of the comparisons are troubling. On L365, I have no idea what values of MSE would be high or low. Case A certainly performs better than cases B–E. However, all the comparisons confirm is that your NN did what it was designed to do (estimate effective radius from the outputs of a Mie calculation). It doesn't show that your algorithm is any good.

  On L367, I disagree that case A is superior. It has the highest Pearson coefficients, but it's results also show the most structure, with some manner of saturation apparent as the Angstrom exponent exceeds 0.5. This is not evident in cases B–D.

- Figs. 1 and 5 convey very little and seem drawn from a poster on this topic. Figs. 6–13 are appallingly bad. The font is too small, there are too many virtually identical plots for the length of discussion, and (considering the large number of data points indicated in §2.4) should probably be density histograms rather than scatter plots to illustrate the spread of data. At the moment, this algorithm looks fairly bad as the spread is large compared to my understanding of the typical uncertainties in existing techniques. The $R^2$ values indicate a dense packing of points near the one-to-one line that none of these plots makes evident.

Various more minor points and comments:

- Please refer to the cases of Tab. 2 uniformly by either letter or $\alpha/\beta$ sets; I prefer the brevity of the letters.

L204 Fig. 3 implies you used more than five input neurons. Please clarify.

L221 Possibly remind the reader that the min/max values are given in Fig. 3.

L230 to L233 doesn't actually explain why you picked the $\chi^2$ test; you just state it's "good". Either write an actual explanation or omit these sentences (as the $\chi^2$ test is fairly common).

§2.3 A mostly unnecessary summary of basic statistical concepts.

L266 The values in Tab. 2 indicate you consider virtually no coarse mode aerosol, which I'd define as at least a few microns. This seems a rather important omission.

L276 to L294. Why do you spend a whole paragraph telling us you shuffled the data? If shuffling is so important, you're going to need to do more than assure me that doing so avoids biases. If it's not, there's no need to mention it.

L306 This isn't a sentence and I don't know if you mean that the ANN was optimised for retrieval of effective radius or complex refractive index. The remainder of the document implies the former.

L309 Can an ANN account for a priori at all?

Fig. 6 This, and subsequent, are scatter plots not correlation plots.

§3.2 The loss of sensitivity with Angstrom exponent is interesting. Both it and lidar ratio are understood as measures of particle size, but the differing sensitivities implies otherwise.

L351 If you aren't going to show us any results from previous algorithms, you need to summarise their results and explain why yours are equivalent.

L354 Is it fair to compare analyses using three observations to four this way?

L382 You are the first to show such strong correlations because you were the first to look, not because of any relative skill in your technique.

L383 Could you explain the reasoning behind this supposition?

L422 These numbers should be one or two significant figures.

L669 Why are you mentioning work not done in a figure caption? That should be in the main text instead. Also, that flow chart implies you select the lowest MSE not by evaluating a number of circumstances but rather by considering it's absolute value. If this is the case, 'lowest' is an extremely poor choice of word.

The English has a peculiar grammar and is overly verbose but was no worse than I usually encounter from non-native speakers. I felt I understood the authors throughout the paper. The technical corrections I caught while reading follow:

L14 I think the following may be preferable: ANN could be a useful alternative or supplement to the traditional approach . . .

L17 allows for investigation of the information

L37 and at the ground are

L40 uncertainty in climate changes forecasts with regard

L43 They appear at various heights in the atmosphere

L54 data acquisition are far less

L64 amount of data, the addition of vertical

L70 reliable, quality-assured instruments and software for data analysis. Automated algorithms

[Figure]

L82  late 1990s to the mid 2000s

L96  results of a large-scale analysis

L97  during a 5-year effort

L98  used for support of data inversion

L99  of the human brain

L104  is unlike that in statistics

L107  area, in part because of the lack of appropriate

L122  among which the most

L186  I'm not certain what this sentence means. My best guess is, "This number of neurons was selected because it resulted in the minimal Mean Squared Error (MSE) in the training phase."

L200  the more difficult it becomes

L203  I'm not certain what the end of this sentence is getting at. My best guess is, "We find that five hidden neurons proves a reasonable compromise between the complexity of the NN, the effort required to train it, and the ability to enhance the network in future studies."

L216  I think 'potential' would be preferable to 'possible' as the former means "any rows that may exist in our case" while the latter means "any rows that could exist in any study", though neither word is actually necessary here.

L227  Shouldn't Pearson's $r$ be italicised?

L229  at the end of the results

L239 lines and identify the slope

L245 The typesetting of this equation could be improved (for example, make the root symbol larger than the subsequent terms).

L263 Either 'for which we need at least 3 hours computation time' or 'for which we need more than 3 hours computation time'.

L299 Either 'a number of' or 'various'; no need to use both.

L361 data. However, extinction profiles

L370 Doesn't this sentence repeat the previous?

L390 real part are more accurate than the imaginary part

L396 the accuracy of outputs depends on

L423 This result suggests that

---

## Referee Comment (RC3) · Anonymous Referee #1 · 1 Apr 2016

The topic of this manuscript is of interest for atmospheric science. The language itself is acceptable, but often the descriptions are vague and sometimes different terms are used for the same thing. Furthermore, the advantages and disadvantages of this new approach are not well discussed. I'm afraid that the manuscript is only of limited usefulness in its current state. The paper needs strong revisions before it may become publishable in AMT.

General comments:

1) First, I have to say that I'm not familiar with ANN. Though the authors try to explain

the ANNs, their explanation of ANN in general and how they use ANN for lidar data inversion is not easy to follow and also sometimes confusing. Unfortunately, I was not really able to figure out how exactly the presented calculations were performed. Some specific points are mentioned below. In any case, the description of the methods needs to be improved significantly before this manuscript can be published.

2) Uncertainties: As the uncertainties of some retrieved parameters are (generally from the physical point of view) determined by the measurement uncertainty in combination with the assumed model (here spherical Mie particles), the presented uncertainties have almost nothing to do with the real uncertainties of the retrieved parameters. In addition, ANN provides some kind of black box, and thus it seem unlikely to me that ANN will ever be able to provide physically consistent quantification of uncertainties of retrieved parameters. Despite these limitations and keeping them in mind, ANN applied on lidar data might be useful for operational purposes if it has significant advantages (e.g. with respect to speed) compared to other methods.

Independent of such considerations, it is mandatory for a feasibility study to consider measurement uncertainties because they can lead to a non-feasibilty of the whole approach. In my view, this is a significant gap in this manuscript, which needs to be filled somehow, as long as this study is called feasibility study.

Some specific comments:

What is the mean radius of a log-normal distribution? The average radius? Usually, log-normal distributions are parameterized with a "mode radius" (which is the median radius) and some parameter for the width of the distribution. The mean radius is not the same as the mode/median radius. Please clarify which of these radii you mean. Adding the formula for the log-normal distribution might be useful.

Line 16-17: It remains unclear why ANN would be particularly useful for the investigation of the information content of optical data. A consistent uncertainty treatment would be necessary for that (see also general comment 2).

Line 32: Extinction-based angstroms are only available when 2 alphas are available. So it remains unclear how this and the previous sentence are related.

Line 53: Remove "profiles of".

Line 140-144 are very unclear.

Line 149: The meaning of "... an algorithm, i.e., perceptron named the ..." is unclear.

Replace "true solution" (e.g. line 151) by "true value".

Line 151: "in which the network errors ... again check for new weight values" does not make sense.

Line 157: What means "fixed random bias value"?

Line 161/162 is confusing. I though the activation function is given in Eq. 1 and is not a sum.

Line 173-175: I do not really understand what would be the benefit of using the output value as input. If you know the output value you don't need any algorithm because you already have the value you want.

Line 263: "more than 3 hours at least." Is this really an issue for the training phase?

Line 263: What means "data downscaling"?

Line 289-290: What is the use of subdividing the "training data" in 'training', 'testing', 'validation'? This is not explained. Subdivision of "training data" into 'validation' seems odd by itself. Looking at the tables there seems to be almost no difference in "Rˆ2" between the 'training', 'testing', 'validation' data sets. Please explain why this subdivion is necessary.

Line 462: "For the first time to our knowledge" could be removed.

Line 469: This is the range of "mean radii" (Tab 1.), but your size distributions also contain radii larger than the mean radius.

Line 479-480: Is this really necessary? The data in table 1 shows that you already have size distributions with effective radii larger than 4 mircrometer. Are larger effective radii really relevant for atmospheric science? If yes, do you expect that lidar with a maximum wavelengths of 1064nm would be able to quantify such aerosols?

Table 1: Do you really use 10011 imaginary part steps?

Table 1: I don't understand the meaning of the rows below imaginary part. I though "N" would be the number of cases in certain ranges of reff, real part, and imaginary part, that are defined in the parameter/value lines in the same table. However, there are 198 million combinations defined in the parameter/values lines, but the sum of N for reff is about 1.6 million, for the real part and the imaginary part 0.9 million. What does that mean?

Table 2: Why use "EC"/"BC" if there is already alpha and beta?

Tables: What is the difference between "Rˆ2 value" and "Adjusted Rˆ2"?

Table 3, 4, 5: Values in "Training Statistics, Rˆ2 value Training", "Training Statistics, Rˆ2 value Testing", "Training Statistics, Rˆ2 value Validation", "Simulation statistics, Pearson's r" are almost the same for each input combination. In particular, can you explain why the "Pearson's r" values are almost the same as the "Rˆ2" values?

Figure 1: There are several arrows that don't make sense. For example, what is the meaning of the arrow from "ANN topology selection" to "Five basic data combinations"?

Figure 1: Results shouldn't be included in a flow chart, references to tables, figures, sections are sufficient.

Figure 3: Why use microphysical properties as input? This doesn't make sense in my view.

Figure 3: Add units.

Figure 4: What is the meaning of "1" in the circles?

Figure 5: Why are there numbers in this plot?

Figure 6-13: Try to reduce the number of subplots and try to make them larger.

Table 10: The network architecture for ANN #10 should be "traingdx" according to the text.

It seems that "ANN" and "NN" are used interchangably in most parts of the paper. I suggest to use only a single abbreviation.